# Application of Exogenous Melatonin Improves Tomato Fruit Quality by Promoting the Accumulation of Primary and Secondary Metabolites

**DOI:** 10.3390/foods11244097

**Published:** 2022-12-19

**Authors:** Jianhua Dou, Jie Wang, Zhongqi Tang, Jihua Yu, Yue Wu, Zeci Liu, Junwen Wang, Guangzheng Wang, Qiang Tian

**Affiliations:** 1College of Horticulture, Gansu Agriculture University, Lanzhou 730070, China; 2State Key Laboratory of Aridland Crop Science, Gansu Agricultural University, Lanzhou 730070, China

**Keywords:** melatonin, tomato fruit, sugars, organic acids, amino acids, phenolic acids and flavonoids, E-nose, sucrose metabolism

## Abstract

Melatonin plays key roles in improving fruit quality and yield by regulating various aspects of plant growth. However, the effects of how melatonin regulates primary and secondary metabolites during fruit growth and development are poorly understood. In this study, the surfaces of tomato fruit were sprayed with different concentrations of melatonin (0, 50, and 100 µmol·L^−1^) on the 20th day after anthesis; we used high-performance liquid chromatography (HPLC) and liquid chromatography/mass spectrometry (LC/MS) to determine the changes in primary and secondary metabolite contents during fruit development and measured the activity of sucrose metabolizing enzymes during fruit development. Our results showed that 100 µmol·L^−1^ melatonin significantly promoted the accumulation of soluble sugar in tomato fruit by increasing the activities of sucrose synthase (SS), sucrose phosphate synthase (SPS), and acid convertase (AI). The application of 100 µmol·L^−1^ melatonin also increased the contents of ten amino acids in tomato fruit as well as decreased the contents of organic acids. In addition, 100 µmol·L^−1^ melatonin application also increased the accumulation of some secondary metabolites, such as six phenolic acids, three flavonoids, and volatile substances (including alcohols, aldehydes, and ketones). In conclusion, melatonin application improves the internal nutritional and flavor quality of tomato fruit by regulating the accumulation of primary and secondary metabolites during tomato fruit ripening. In the future, we need to further understand the molecular mechanism of melatonin in tomato fruit to lay a solid foundation for quality improvement breeding.

## 1. Introduction

Melatonin is an indole molecule widely found in plants and animals; as an endogenous metabolite of living organisms, it can be degraded by the organisms themselves and is non-toxic, safe, and residue-free to humans [1]. One notable difference in melatonin synthesis in animals and plants concerns the availability of the synthetic precursor tryptophan. Unlike plants, animals need to obtain tryptophan from their diet [2]. Melatonin can be considered a biostimulant molecule; the use of melatonin in agriculture could be an alternative and sustainable method to the use of agrochemicals [3,4]; melatonin also has a positive potential in phytostimulatory activities [5] and genetic breeding [6]. A large number of studies have shown that melatonin has a powerful free radical scavenging ability to enhance plant resistance to various biotic and abiotic stresses and its regulative role in regulating plant signaling and response pathways [7]. For example, relatively low melatonin concentrations enhanced plant stress resistance, such as drought [8], salt [9], low temperature [10,11], high temperature [12], and heavy metal stress [13,14]. In addition, several studies have reported that melatonin can also regulate the appearance, quality, and nutrition of the fruit. For instance, the contents of starch, sucrose, fructose, glucose, and protein in soybean treated with melatonin significantly increased [15]. After irrigating the exogenous melatonin solution [16], the contents of ascorbic acid, lycopene, soluble solids, and phosphorus in tomato fruit were found to be enhanced compared to the control [17]. Exogenous melatonin application increases soluble sugar (especially sucrose and sorbitol) content during pear fruit ripening [18]. Exogenous melatonin application significantly increased the soluble solids and phenolic content, thus improving the quality of strawberries [19]. Melatonin treatment significantly up-regulated *SlADH2* and *SlAAT* involved in aromatic substance synthesis in tomato fruit, thereby increasing postharvest tomato fruit aroma substance content [20]. Taken together, the results showed that melatonin is involved in plant growth and stress resistance; moreover, it has the potential role to improve fruit quality. These features make melatonin a commercially important ingredient in modern agriculture as a growth regulator for cash crops to improve crop yields and ensure food safety.

Tomato (*Solanum lycopersicum*) is a widely cultivated fruit or vegetable crop worldwide [21], which has an essential function in restructuring agriculture and increasing the income of farmers. In terms of nutrients, tomatoes contain many health-promoting compounds, mainly including sugars (sucrose, glucose, fructose) [22], carotenoids (lycopene, beta-carotene) [23], phenols (flavonoids and phenolic acids) [24], amino acids [25], vitamin C and E, minerals [26], etc., which are essential nutrient components of a balanced diet. There was a significant correlation between the internal nutrition and flavor of tomato fruit and the level of consumer preferences [27]. With the improvement of people’s living standards and the emphasis on health, there is a rapidly increasing demand for quality fruit and vegetables [28]. However, the difference in tomato nutrients and flavors can be attributed to various factors, such as cultivars [29], environmental factors [30], agronomic practices [31], and metabolic regulation [32]. Thus, how to efficiently improve the quality of tomato fruit has been a key area of research.

Previous studies on melatonin have focused on the stress resistance of plants or postharvest quality and preservation of fruit. However, there are few reports on the effects of melatonin on primary and secondary metabolites during fruit growth and development. The flavor and nutrition in tomato fruit are generally reflected by the content and composition of primary metabolites (including organic acids, soluble sugars, and amino acids) and secondary metabolites (including flavonoids and phenolic acids) during the development of the fruit [33,34]. Consequently, the primary and secondary metabolites of tomato fruit sprayed with exogenous melatonin were determined in this research. Furthermore, the composition of soluble sugars and the activities of critical enzymes related to sucrose metabolism were also explored to understand the regulative effects of melatonin. Our study provides theoretical support for the application of melatonin in tomato fruit quality improvement.

## 2. Materials and Methods

### 2.1. Materials and Experimental Design

Tomato (*Solanum lycopersicum* cv. 184) plants were grown in a solar greenhouse in Yuzhong County, Lanzhou City, China (35.85° N, 104.09° E).

During the experimental treatments, we labeled 20 healthy plants per treatment and repeated them three times. The second fruit cluster was selected for treatment. When the second spike of flowers bloomed, the flowering date was marked. For each treatment to ensure equal fruit maturity, 2–3 fruit with the same date of anthesis were chosen for each plant. The fruit was treated as follows: control (deionized water, CK, containing 0.1% Tween-20 and 0.1% ethanol), an aqueous solution of melatonin (containing 0.1% Tween-20 and 0.1% ethanol) at two concentrations (50 µmol·L^−1^, T1; 100 µmol·L^−1^, T2). During treatment, every fruit was sprayed until the solution dripped. The fruit was treated on the 20th day after anthesis; the treatments carried on at 6-day intervals until the fruit ripened. Moreover, samples of tomato fruit were obtained at 6-day intervals during the treatment period until tomato fruit ripening. For sampling, five fruit from each treatment were randomly selected and replicated three times. The samples we obtained were immediately well frozen in liquid nitrogen and then stored at −80 °C for the indicators to be measured.

### 2.2. Sugar Components

The 5.0 g homogenate of tomato was transferred to a 50 mL centrifuge tube and fixed to 20 mL with ultrapure water. The homogenate was sonicated in a water (30 °C) bath for 60 min and then centrifuged for 10 min (4 °C, 10,000 r·min^−1^). The supernatant (2 mL) was aspirated and passed through a disposable aqueous membrane with a 0.22 µm, which was used for high-performance liquid chromatography (HPLC). The HPLC conditions were based on a method by Wang et al. [35]. The chromatographic column was an LC-NH2 amino column (250 mm × 4.6 mm, Waters Corp., Milford, MA, USA), the mobile phase used was V (acetonitrile): V (water) = 3:1 and its flow rate was 1.0 mL·min^−1^, the column temperature was 30 °C, and the injection volume was 20 µL.

### 2.3. Organic Acid Components

The organic acid components were analyzed according to Wang et al., with minor modifications [27]. First, 5.0 g fresh and clean fruit samples were transferred into a 50 mL centrifuge tube, fixed to 25 mL with ultrapure water, and homogenized by centrifugation for 10 min. Then, 1 mL of the supernatant was aspirated and filtered through a disposable aqueous membrane with a 0.22 µm into a 2 mL brown sample bottle, and the filtrate was used for HPLC. The HPLC conditions were based on a method by Wang et al. [35]. The chromatographic column was Hi-PiexH (300 mm × 7.7 mm), the mobile phase used was sodium dihydrogen phosphate at a concentration of 0.2 mmol·L^−1^, and its flow rate was 0.5 mL·min^−1^, the column temperature was 30 °C, and the injection volume was 5 µL.

### 2.4. Amino Acid Components

We used liquid chromatography/mass spectrometry (LC/MS) to quantify the amino acids in tomato fruit [36]. Extraction of lyophilized fruit powder (0.1 g) with 0.5 mol·L^−1^ hydrochloric acid solution (1 mL) in a 2 mL centrifuge tube. The tube was vortexed and mixed for 20 min, sonicated in a water (25 °C) bath for 20 min, and then centrifuged at 20,000× *g* for 20 min. Finally, 250 µL of supernatant was moved to a brown sample bottle, and then 750 µL of 80% acetonitrile aqueous solution was added. The HPLC conditions were based on a method by Wang et al. [36]. The mass spectrometry conditions were as follows: ionization mode was electrospray ionization in positive ion mode; gas flow rate was 13.0 L·min^−1^; the dry gas temperature was 330 °C; nebulizer was 35 psi; sheath gas flow rate was 12 L·min^−1^; sheath gas temperature was 390 °C; nozzle voltage was 0 V, and capillary voltage was 1500 V.

### 2.5. Phenolic Acids and Flavonoids Components

The extraction and determination of the phenolic acids and flavonoid content were based on a previously optimized method in our laboratory [37]. First, lyophilized samples were weighed (0.1 g) and homogeneously extracted for 1 h in 2 mL methanol, with constant shaking during extraction to allow full extraction. Extraction of the solution was centrifuged at 8000 rpm and continued for 10 min; the supernatant was removed and passed through a disposable filter membrane into a brown bottle awaiting determination. The HPLC conditions were referred to the method of Jin et al. [37], a symmetrical C18 column (250 mm × 4.6 mm, 5 µm, Waters Corp., Milford, MA, USA), the column temperature was 30 °C, sample volume was 10 µL, mobile phase A was methanol, mobile phase B was 1% (*v*/*v*) acetic acid, and the flow rate was 1.1 mL·min^−1^. Sixteen different compounds were detected at different wavelengths (including 322, 280, and 240 nm) (Table 1).

### 2.6. Volatile Content by Electronic Nose Analysis

The PEN3 E-nose (Airsense Analytics GmbH, Schwerin, Germany) was equipped with ten sensors. Ten sensor signal response expressed as (G/G0), the ratio of the conductivity of volatile substances to that of pure air [38]. Different sensors can detect different odor-sensitive substances. Table 2 lists our used sensors in terms of substance type and performance description. The electronic nose detection of tomato fruit was based on a previously optimized method in our laboratory, with a slight modification [39]. The tomato homogenate (3.0 g) was accurately weighed into a 20 mL brown headspace sample bottle, sequentially added 1.5 g of sodium chloride and a stirring magnetic rotor, and quickly tightened the cap of the bottle. We equilibrated the gas in the bottle by heating it on a thermostatic magnetic stirrer (50 °C) for 15 min, and finally, the detector needle was inserted into the sample bottle to measure the odor-sensitive substances. Instrument detection parameters were as follows: sensor flush: 60 s, sensor zeroing: 10 s, and measurement time: 180 s. All samples were repeated three times.

### 2.7. Activities of Enzymes Related to Sucrose Metabolism

The activities of SS, SPS, AI, and NI were measured using ELISA kits from Quanzhou Ruixin Biotechnology Co., Ltd., Quanzhou, China). First, fresh samples were accurately weighed (0.5 g) and well-ground in liquid nitrogen. Then, a wash with 4.5 mL of 0.01 mol/L PBS (pH: 7.2–7.4) into a centrifuge tube. We centrifuged the tubes for 30 min (8000 rpm, 4 °C) and collected the supernatant. After that, the enzyme extract and reagents were added to the enzyme plate in order according to the instructions of the kit. Finally, we measured the absorbance value at 450 nm using the HBS-1096A microplate reader (Detie, Nanjing, China) and calculated the concentration from it.

### 2.8. Statistical Analysis

Microsoft Excel 2010 (Microsoft Inc., Redmond, WA, USA) was used to make all tables, and Origin 2022 (Origin, Inc., San Francisco, CA, USA) was used to make figures. We used one-way ANOVA and Duncan’s multiple range test (*p* < 0.05) in SPSS (version 22.0; SPSS Institute Inc., Chicago, IL, USA) software for data analysis.

## 3. Results

### 3.1. Effects of Exogenous Melatonin on Sugar Components in Tomato Fruit

Fructose, glucose, and sucrose contents of tomato fruit were measured 20–52 d after anthesis. The levels of fructose gradually increased with the growth and development of tomato fruit (Figure 1A), which reached the highest level at 52 d. We can also see that there was no significant difference in fructose content among treatments at 24–32 d, and the fructose content of melatonin-treated fruit was higher compared to control fruit at 48–52 d. At 52 d, the fructose content of the T2 treatment increased by 24.26% compared to the control. Glucose content also increased with fruit ripening (Figure 1B). During 34–52 d, the glucose content of melatonin-treated fruit was significantly higher compared to the control and reached the maximum at 52 d. The content of glucose in T1 and T2 treatments increased by 17.78% and 23.67%, respectively, compared to the control. Sucrose content was relatively high at the beginning of fruit development (20 to 27 d) and then gradually decreased during fruit ripening (Figure 1C). Levels of sucrose were significantly higher in fruit treated with melatonin than in the control group at other time points.

### 3.2. Effects of Exogenous Melatonin on Organic Acid Components in Tomato Fruit

Different exogenous melatonin affected the organic acid content of tomato fruit. The quinic acid content first increased and then decreased during fruit growth and development (Figure 2A). The content of quinic acid reached the maximum at 41 d. Then at 52 d, the quinic acid content of fruit in T2 treatment was significantly lower than the control. The tartaric acid content showed a continuous decreasing trend during fruit ripening (Figure 2B). No significant difference in the content of tartaric acid between the treatments was observed from 20 to 34 d. Moreover, at 52 d, the content of tartaric acid in T2 treatment was decreased by 7.42% compared to that of the control. During the fruit ripening process, the content of malic acid in all treatments was first increased and accumulated before 34 d and then decreased to the lowest at 52 d (Figure 2C). It was observed that T2 treatment significantly increased the malic acid content by 24.32% compared to the control at 52 d. The citric acid content in fruit showed an overall pattern of increasing and then decreasing with the ripening (Figure 2D). From 20 to 34 d, the content of citric acid in fruit showed no difference among treatments, but its content in the T2 treatment was clearly lower than the control group at 48–52 d.

### 3.3. Effects of Exogenous Melatonin on Amino Acid Components in Tomato Fruit

The content of 16 amino acids (threonine, lysine, phenylalanine, tryptophan, leucine, isoleucine, valine, methionine, cysteine, arginine, glutamic acid, tyrosine, aspartic acid, serine, glycine, and alanine) was determined and analyzed by LC-MS.

Among the essential amino acids, threonine and phenylalanine content were the highest, and methionine content was the lowest (Table 3). Tryptophan content continued to increase during fruit ripening, while isoleucine and valine content was relatively high at the beginning of fruit development and then kept decreasing. At 52 d, the fruit content of lysine, threonine, tryptophan, phenylalanine, isoleucine, leucine, and valine in the T2 treatment was significantly higher than that of the control. Moreover, there was no significant difference in methionine.

Glutamic acid increased significantly in the fruit during ripening, making it the most abundant amino acid in ripe fruit, followed by aspartic acid, and the lowest was glycine (Table 4). After 52 d, the contents of glutamic acid, tyrosine, aspartic acid, and glycine were significantly higher in T2-treated fruit than in control, while cysteine, arginine, and serine were not significantly different. Melatonin treatment had no significant effect on cysteine content from 20 to 27 d. However, the cysteine content of the T2 treatment increased at 34 d, which was significantly higher than the control group.

### 3.4. Effects of Exogenous Melatonin on Phenolic Acids and Flavonoids in Tomato Fruit

We quantified the content of 4 flavonoids and 12 phenolic acids in tomato fruit by HPLC and performed cluster analysis and unsupervised principal component analysis (PCA), which confirmed the high reproducibility among the three biological replicates and treatments (Figure 3). In Figure 3, the classification model for the parameters of the principal component analysis of tomato fruit polyphenolic compounds under melatonin treatment is shown. The first and second principal components of PC1 (45.8%) and PC2 (19.1%) for melatonin treatment explained 64.9% of the total variance (Figure 3A). In addition, the loadings plots showed that naringin and sinapic acid had high first principal component loadings, while 4-coumaric acid and p-hydroxybenzoic acid had high second principal component loadings (Figure 3B). The CK, T1, and T2 treatments produced a visible separation based on PC1, which was confirmed by the results of the clustering analysis (Figure 3C). To further clarify the polyphenol metabolites of melatonin-treated tomatoes, we performed a hierarchical cluster analysis (HCA). The heat map included data for 12 phenolic acid and four flavonoid metabolites identified in tomato fruit (Figure 3C). The T2-treated group contained higher levels of three flavonoids (rutin, quercetin, naringenin) and six phenolic acids (caffeic acid, p-hydroxybenzoic acid, protocatechuic acid, chlorogenic acid, gentisic acid, and sinapic acid). The T1-treated group contained higher levels of cinnamic acid, gallic acid, and kaempferol. Appendix A shows the data on the content of polyphenolic components.

### 3.5. Effects of Exogenous Melatonin on Volatile Flavor Intensity in Tomato Fruit

As shown in Figure 4A, the radar map revealed the differences in the contribution values of the ten sensors of the electronic nose to the sensitive material substances in the tomato fruit under different melatonin treatments. We could have seen that the W2W, W3S, and W5S sensors had higher relative resistivity values compared to the other sensors. In addition, the clustering thermogram results further showed that melatonin treatment altered the aroma composition of tomato fruit, where the response values of most sensors were increased under T2 treatment compared to control, corresponding to volatiles such as organic sulfides (W2W), hydrides (W6S), aldehydes, ketones, and alcohols (W2S), short chain alkanes aromatic fractions (W5C), and aromatic benzene (W1C) (Figure 4B). Appendix A shows the data from the electronic nose.

### 3.6. Effects of Exogenous Melatonin on Sucrose Metabolism-Related Enzyme Activities in Tomato Fruit

The activities of the enzymes associated with sucrose metabolism are shown in Figure 5. SS and SPS had relatively high activities at the beginning of fruit development, which gradually decreased with maturation (Figure 5A,B). Melatonin-treated fruit showed higher activities of SS and SPS at maturity compared to control fruit. During fruit ripening, the activities of NI and AI revealed a trend of first increasing and then decreasing (Figure 5C,D). There was no significant difference in NI activity at 52 d among the treatments. However, melatonin application of T2 increased the activity of AI in fruit to a great extent at 52 d.

## 4. Discussion

### 4.1. Melatonin Application Increase Sugar Content in Tomato Fruit

The development of taste is due to a universal increase in sweetness as a result of increased sugar isomerism, which includes hydrolysis of polysaccharides (especially starch), decreased acidity, and a suitable sugar-to-acid ratio [40,41]. The sugars in tomato fruit are mainly fructose, glucose, and a small amount of sucrose [42]. Previous studies have shown that exogenous melatonin improves the yield and quality of soybean by promoting the accumulation of photosynthetic products, such as sucrose, starch, glucose, and fructose [15]. In our study, the content of fructose in tomato fruit at 48 d after anthesis was increased by melatonin treatment (Figure 1A) and glucose content at 34 d (Figure 1B), but melatonin treatment increased sucrose content in tomato fruit as early as 27 d (Figure 1C). These results suggest that sucrose may take a predominant role in response to melatonin, although the content of fructose and glucose account for approximately 95% or more of the total soluble sugars in tomato fruit. Therefore, we further analyzed the activities of the enzymes involved in sucrose metabolism. Our results showed that during fruit development (i.e., 34–52 d), melatonin not only promoted the SS activity (Figure 5A) but also the SPS activity (Figure 5B), thereby promoting the biosynthesis of sucrose. NI and AI are key enzymes in sucrose catabolism [43]. Our results indicated that the application of exogenous melatonin during fruit development could significantly increase the activity of AI, which accelerated the hydrolysis of sucrose to fructose and glucose; these results were in accordance with the research on grapes [44]. The above results demonstrate the positive contribution of melatonin to the accumulation and conversion of sugar in tomato fruit.

### 4.2. Melatonin Application Decrease Organic Acid Content in Tomato Fruit

The content of organic acid is a limiting factor for the flavor of tomato and plays an indispensable role in the formation of fruit flavor [45]. The main organic acids in tomato fruit are citric acid and malic acid [46]; meanwhile the fruit also contains small amounts of oxalic acid, quinic acid, tartaric acid, and others [47]. Previous studies have reported that organic acid content increases during tomato ripening [48] or gradually increases during early tomato fruit development and then decreases [49]. Our study of organic acids showed that citric acid content was dominant throughout the development of tomatoes (Figure 2D), while malic acid content was the highest at 34 d and then decreased at fruit ripening (Figure 2C); these results were similar to the results of Agius et al. [50]. In addition, exogenous melatonin application significantly reduced citric (Figure 2D), tartaric (Figure 2B), and quinic acid (Figure 2A) contents and increased malic acid content at 52 d (Figure 2C). The above results indicated that melatonin significantly reduced the acidity of tomato fruit. Combined with the results of the sugar component, our results indicated that melatonin could improve the flavor quality of tomato fruit.

### 4.3. Melatonin Application Regulate Amino Acid Content in Tomato Fruit

The amino acid composition is one of the most important indicators for evaluating the nutrition and flavor of tomato fruit [25,51], which are not only involved in protein biosynthesis but also significantly associated with taste perception in humans [52]. Studies have shown that amino acids increase sharply during fruit ripening, and their abundance changes differently. In general, glutamic acid, glutamine, GABA, and aspartic acid in tomato fruit exhibit relatively high levels throughout ripening, accounting for approximately 80% of the total free amino acids in ripe tomato fruit [53,54]. In this study, the fruit of T2 treatment had relatively higher contents of lysine, threonine, phenylalanine, tryptophan, leucine, isoleucine, valine, tyrosine, aspartic acid, and glycine than the CK after 52 d. Glutamic acid is one of the most abundant amino acids in tomato fruit, and it has a typical “umami” flavor [55]. Our results showed that exogenous melatonin significantly increases the glutamate content compared with the control group, and this may be partly due to the stimulation of glutamate dehydrogenase. Tryptophan, phenylalanine, and tyrosine are aromatic amino acids involved in the shikimic acid pathway [56], which play an important role in the aroma development of fruit [57,58]. Our results indicated that the application of exogenous melatonin significantly increases the aromatic amino acid (including tyrosine, tryptophan, and phenylalanine) content at 52 d. These results suggest that melatonin alters the level of amino acid content, which in turn may affect the content of volatile substance components in tomato fruit.

### 4.4. Melatonin Application Regulate Secondary Metabolites Content in Tomato Fruit

The secondary metabolites phenolic acids and flavonoids have an extensive range of physiological properties, including anti-allergic, anti-inflammatory, antibacterial, and antioxidant effects [59,60], which are all important compounds of tomato fruit for human health [61]. Previous research has shown that the application of melatonin significantly enhanced the accumulation of total phenols in barley (*Hordeum vulgare* L.) [62] and increased the level of phenolic compounds in jujube (*Ziziphus jujuba* Mill.) fruit by triggering the phenolic biosynthesis pathway [63]. Our results showed that T2 treatment enhanced the accumulation of six phenolic acids and three flavonoids in fruit. These further suggest that the impact of exogenous melatonin on the improvement of tomato fruit quality is significantly related to its promotion of the accumulation of bioactive substances, such as flavonoids and phenolic acids compounds [64,65,66].

### 4.5. Melatonin Application Increases the Volatile Substance Content in Tomato Fruit

Volatile metabolites biosynthesized during tomato ripening can influence the flavor and aroma of the fruit [67,68]. Previous studies have shown that wines made from melatonin-treated grape berries have stronger sensory characteristics of fruitiness, spiciness, and sweetness [69]. Our results showed that melatonin had improved the content of volatile substances such as ketones, short-chain alkane aromatic fractions, alcohols, aldehydes, and aromatic benzene in the fruit, thereby enhancing the fruit flavor of tomatoes. Up to date, researchers have identified over four hundred volatile compounds in tomato fruit, of which only approximately thirty of them (hexanal, cis-3-hexenal, trans-2-hexenal, etc.) were considered to be characteristic volatiles that contributed to tomato flavor [70,71]. However, the specific characteristic substance that responded to the regulation of melatonin and influenced the fruit flavor in this study still needs to be further tested and analyzed.

## 5. Conclusions

This study showed that melatonin spraying during fruit growth could regulate the enzymatic activities (SS, SPS, AI) in sucrose metabolism and had positive effects on the accumulation of the soluble sugars in tomato fruit. In addition, exogenous melatonin regulated the primary metabolites during tomato fruit ripening, such as decreased content of organic acid and increased content of most amino acids. Exogenous melatonin also regulates secondary metabolites, such as six phenolic acids, three flavonoids, and volatile substances (including alcohols, aldehydes, and ketones). In conclusion, melatonin is systematically involved in the regulation of primary and secondary metabolism, which improves the nutritional quality and flavor quality of tomato fruit. In the future, we need to further understand the molecular mechanism of melatonin in tomato fruit to lay a solid foundation for quality improvement breeding.

## Figures and Tables

**Figure 1 foods-11-04097-f001:**
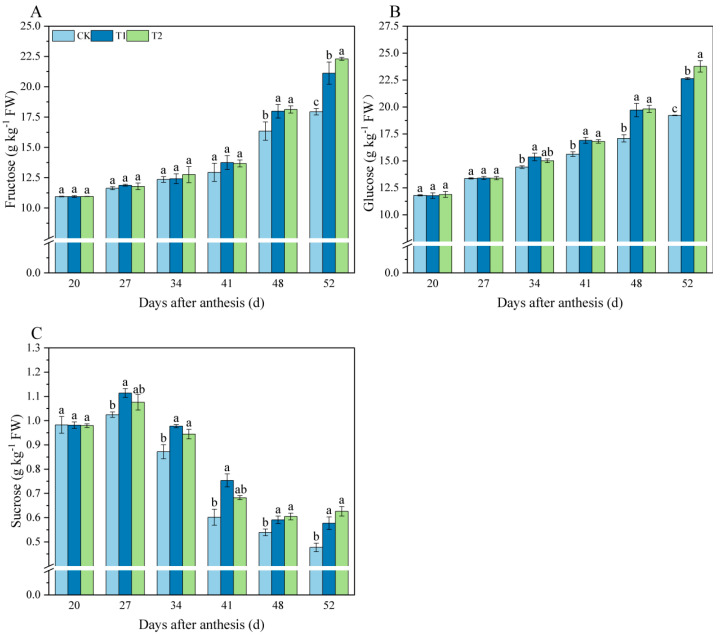
Effects of exogenous melatonin on sugar components content in tomato fruit. (**A**) Fructose. (**B**) Glucose. (**C**) Sucrose. Results are means ± SE of three independent replications. Different letters represent significant differences between treatments (*p* < 0.05).

**Figure 2 foods-11-04097-f002:**
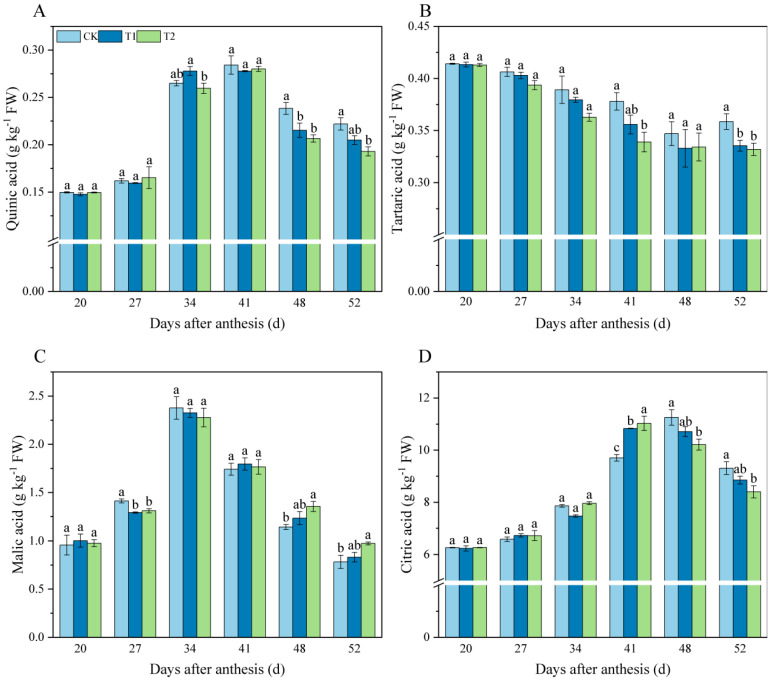
Effects of exogenous melatonin on organic acid components content in tomato fruit. (**A**) Quinic acid. (**B**) Tartaric acid. (**C**) Malic acid. (**D**) Citric acid. Results are means ± SE of three independent replications. Different letters represent significant differences between treatments (*p* < 0.05).

**Figure 3 foods-11-04097-f003:**
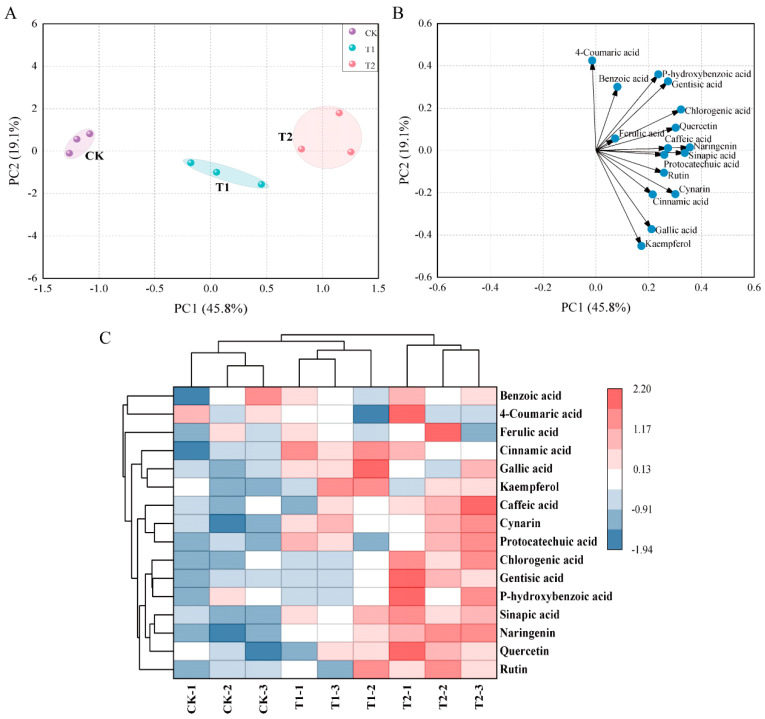
Heatmap Statistical analysis of bioactive metabolites of exogenous melatonin in tomato fruit at 52 d after anthesis. (**A**) PCA scatter plots of phenolic components are shown. (**B**) PCA loadings of phenolic components are shown. (**C**) Heat map of phenolic substance in tomato fruit. The colored areas correspond to the concentration of phenolics in each treatment. Each column represents a treatment, and each row represents a phenolic substance.

**Figure 4 foods-11-04097-f004:**
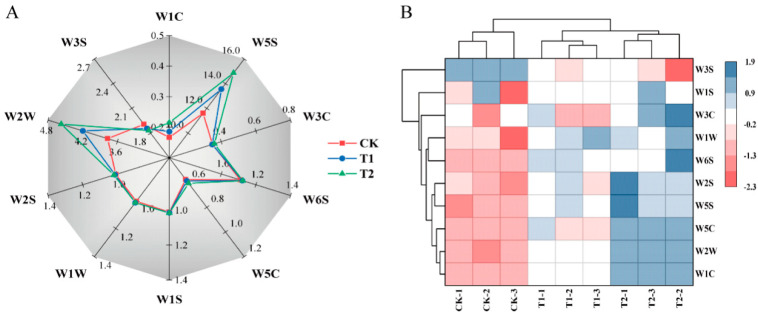
Effects of exogenous melatonin on odor characteristic contribution values (**A**) and hierarchical cluster analysis (**B**) in tomato fruit at 52 d after anthesis. Results are expressed as mean values of three independent replications.

**Figure 5 foods-11-04097-f005:**
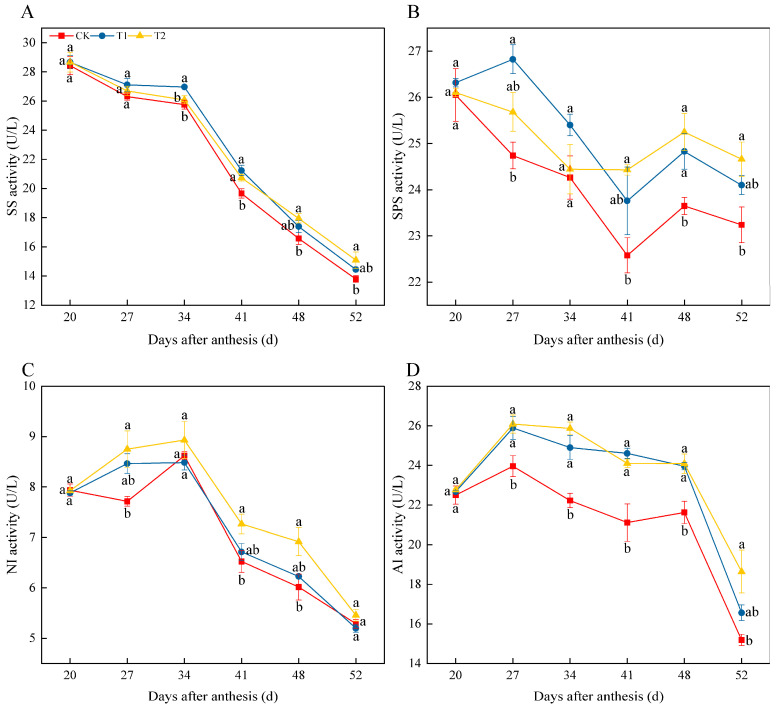
Effects of exogenous melatonin on the activities of enzymes in sucrose metabolism in tomato fruit. (**A**) The activity of sucrose synthase (SS). (**B**) The activity of sucrose phosphate synthase (SPS). (**C**) The activity of neutral invertase (NI). (**D**) The activity of acid invertase (AI). Results are means ± SE of three independent replications. Different letters represent significant differences between treatments (*p* < 0.05).

**Table 1 foods-11-04097-t001:** Detection of different polyphenolic components by HPLC at different wavelengths.

Wavelength	240 nm	280 nm	322 nm
Polyphenolic compounds	Rutin	Gallic acid	Gentisic acid
	Protocatechuic acid	4-coumaric acid	Caffeic acid
	Quercetin	Cinnamic acid	Cynarin
	Chlorogenic acid	Benzoic acid	Sinapic acid
	P-hydroxybenzoic acid	Ferulic acid Naringenin	Kaempferol

**Table 2 foods-11-04097-t002:** Description of the types and properties of substances represented by the ten sensors of the E-nose.

Array Serial Number	Sensor Name	Substance Types	Sensor Performance Description
1	W1C	Aromatic	Aromatic components, benzenes
2	W5S	Broadrange	High sensitivity, sensitive to nitrogen oxides
3	W3C	Aromatic	Sensitive aromatic components, ammonia
4	W6S	Hydrogen	Mainly selective to hydride
5	W5C	Arom-aliph	Aromatic components of short-chain alkanes
6	W1S	Broad-methane	Sensitive to methyl groups
7	W1W	Sulfur-organic	Sensitive to sulfides
8	W2S	Broad-alcohol	Sensitive to aldehydes, alcohols, ketones
9	W2W	Sulph-chlor	Aromatic components, sensitive organic sulfides
10	W3S	Methane-aliph	Sensitive to long-chain alkanes

**Table 3 foods-11-04097-t003:** Effects of exogenous melatonin on the content of essential amino acids in tomato fruit.

Days after Anthesis (d)	Treatments	Essential Amino Acids (mg·g^−1^ DW)							
		Lysine	Threonine	Phenylalanine	Tryptophan	Leucine	Isoleucine	Valine	Methionine
20	CK	1.083 ± 0.031 ^a^	2.393 ± 0.064 ^a^	2.391 ± 0.053 ^a^	0.396 ± 0.002 ^a^	0.697 ± 0.004 ^a^	0.715 ± 0.016 ^a^	1.385 ± 0.007 ^a^	0.128 ± 0.002 ^a^
	T1	1.071 ± 0.007 ^a^	2.461 ± 0.003 ^a^	2.448 ± 0.004 ^a^	0.391 ± 0.002 ^a^	0.644 ± 0.001 ^b^	0.697 ± 0.004 ^a^	1.386 ± 0.001 ^a^	0.129 ± 0.001 ^a^
	T2	1.067 ± 0.002 ^a^	2.463 ± 0.000 ^a^	2.450 ± 0.000 ^a^	0.388 ± 0.005 ^a^	0.692 ± 0.004 ^a^	0.693 ± 0.007 ^a^	1.386 ± 0.000 ^a^	0.129 ± 0.000 ^a^
27	CK	1.029 ± 0.002 ^a^	2.315 ± 0.012 ^c^	2.318 ± 0.017 ^b^	0.432 ± 0.019 ^a^	0.796 ± 0.006 ^a^	0.814 ± 0.008 ^a^	1.312 ± 0.013 ^a^	0.133 ± 0.001 ^a^
	T1	1.008 ± 0.048 ^a^	2.469 ± 0.000 ^a^	2.451 ± 0.003 ^a^	0.441 ± 0.004 ^a^	0.797 ± 0.001 ^a^	0.809 ± 0.001 ^a^	1.250 ± 0.006 ^b^	0.114 ± 0.002c
	T2	1.090 ± 0.005 ^a^	2.387 ± 0.023 ^b^	2.368 ± 0.034 ^b^	0.442 ± 0.001 ^a^	0.798 ± 0.000 ^a^	0.809 ± 0.000 ^a^	1.303 ± 0.016 ^a^	0.122 ± 0.001 ^b^
34	CK	1.053 ± 0.017 ^a^	3.082 ± 0.004 ^a^	3.099 ± 0.012 ^a^	0.477 ± 0.001 ^a^	0.813 ± 0.004 ^a^	0.825 ± 0.007 ^a^	1.320 ± 0.004 ^b^	0.146 ± 0.001 ^a^
	T1	1.092 ± 0.011 ^a^	3.036 ± 0.009 ^b^	3.037 ± 0.002 ^b^	0.485 ± 0.013 ^a^	0.840 ± 0.005 ^a^	0.847 ± 0.004 ^a^	1.361 ± 0.003 ^a^	0.135 ± 0.001 ^b^
	T2	1.021 ± 0.034 ^a^	3.020 ± 0.009 ^b^	3.016 ± 0.002 ^b^	0.497 ± 0.013 ^a^	0.839 ± 0.014 ^a^	0.818 ± 0.034 ^a^	1.238 ± 0.015 ^c^	0.125 ± 0.001 ^c^
41	CK	1.088 ± 0.020 ^b^	3.319 ± 0.168 ^b^	3.272 ± 0.175 ^b^	0.498 ± 0.009 ^b^	0.724 ± 0.043 ^b^	0.734 ± 0.033 ^c^	0.841 ± 0.055 ^b^	0.095 ± 0.007 ^c^
	T1	1.041 ± 0.012 ^b^	3.526 ± 0.003 ^ab^	3.463 ± 0.004 ^b^	0.509 ± 0.004 ^a b^	0.836 ± 0.006 ^a^	0.847 ± 0.006 ^b^	1.117 ± 0.007 ^a^	0.133 ± 0.001 ^a^
	T2	1.146 ± 0.009 ^a^	3.854 ± 0.063 ^a^	3.858 ± 0.063 ^a^	0.537 ± 0.011 ^a^	0.922 ± 0.018 ^a^	0.921 ± 0.004 ^a^	1.058 ± 0.027 ^a^	0.113 ± 0.002 ^b^
48	CK	1.029 ± 0.011 ^b^	3.004 ± 0.052 ^b^	2.995 ± 0.040 ^b^	0.625 ± 0.014 ^c^	0.485 ± 0.006 ^a^	0.481 ± 0.013 ^a^	0.387 ± 0.005 ^a^	0.130 ± 0.003 ^b^
	T1	1.067 ± 0.042 ^ab^	3.042 ± 0.001 ^b^	3.101 ± 0.009 ^b^	0.679 ± 0.000 ^b^	0.469 ± 0.018 ^a^	0.414 ± 0.000 ^b^	0.283 ± 0.001 ^c^	0.146 ± 0.001 ^a^
	T2	1.121 ± 0.005 ^a^	3.683 ± 0.037 ^a^	3.689 ± 0.050 ^a^	0.752 ± 0.020 ^a^	0.482 ± 0.012 ^a^	0.493 ± 0.010 ^a^	0.333 ± 0.006 ^b^	0.148 ± 0.003 ^a^
52	CK	1.068 ± 0.009 ^b^	2.997 ± 0.008 ^b^	2.990 ± 0.033 ^b^	0.738 ± 0.006 ^b^	0.251 ± 0.003 ^c^	0.339 ± 0.002 ^b^	0.237 ± 0.001 ^b^	0.206 ± 0.001 ^a^
	T1	1.177 ± 0.000 ^a^	2.931 ± 0.064 ^b^	2.933 ± 0.079 ^b^	0.758 ± 0.026 ^b^	0.333 ± 0.006 ^b^	0.329 ± 0.013 ^b^	0.222 ± 0.011 ^b^	0.220 ± 0.011 ^a^
	T2	1.197 ± 0.032 ^a^	3.433 ± 0.031 ^a^	3.441 ± 0.029 ^a^	0.897 ± 0.041 ^a^	0.436 ± 0.005 ^a^	0.436 ± 0.008 ^a^	0.318 ± 0.008 ^a^	0.218 ± 0.007 ^a^

Note: Results are means ± SE of three independent replications. Different letters represent significant differences between treatments (*p* < 0.05).

**Table 4 foods-11-04097-t004:** Effects of exogenous melatonin on non-essential amino acids of tomato fruit.

Days after Anthesis (d)	Treatments	Non-Essential Amino Acids (mg·g^−1^ DW)							
		Cysteine	Arginine	Glutamate	Tyrosine	Aspartic Acid	Serine	Glycine	Alanine
20	CK	1.604 ± 0.130 ^a^	0.363 ± 0.110 ^a^	3.983 ± 0.037 ^a^	0.315 ± 0.009 ^a^	0.910 ± 0.009 ^a^	1.085 ± 0.009 ^a^	0.015 ± 0.002 ^a^	1.687 ± 0.013 ^a^
	T1	1.697 ± 0.104 ^a^	0.343 ± 0.021 ^a^	3.995 ± 0.104 ^a^	0.303 ± 0.007 ^a^	0.913 ± 0.002 ^a^	1.081 ± 0.002 ^a^	0.014 ± 0.000 ^a^	1.688 ± 0.002 ^a^
	T2	1.599 ± 0.019 ^a^	0.335 ± 0.005 ^a^	3.998 ± 0.019 ^a^	0.314 ± 0.004 ^a^	0.914 ± 0.000 ^a^	1.080 ± 0.000 ^a^	0.014 ± 0.000 ^a^	1.689 ± 0.000 ^a^
27	CK	1.852 ± 0.144 ^a^	0.370 ± 0.016 ^b^	3.982 ± 0.030 ^b^	0.344 ± 0.006 ^a^	1.012 ± 0.006 ^a^	0.833 ± 0.011 ^a^	0.011 ± 0.001 ^b^	1.451 ± 0.010 ^a^
	T1	1.883 ± 0.024 ^a^	0.460 ± 0.010 ^a^	4.066 ± 0.004 ^b^	0.347 ± 0.001 ^a^	1.083 ± 0.012 ^a^	0.796 ± 0.003 ^b^	0.015 ± 0.001 ^a^	1.244 ± 0.001 ^b^
	T2	1.885 ± 0.004 ^a^	0.406 ± 0.036 ^ab^	4.188 ± 0.035 ^a^	0.347 ± 0.000 ^a^	1.064 ± 0.085 ^a^	0.788 ± 0.003 ^b^	0.012 ± 0.000 ^b^	1.216 ± 0.006 ^c^
34	CK	1.796 ± 0.052 ^b^	0.360 ± 0.121 ^a^	4.071 ± 0.005 ^c^	0.453 ± 0.005 ^b^	2.674 ± 0.018 ^a^	0.933 ± 0.008 ^a^	0.094 ± 0.001 ^a^	1.371 ± 0.003 ^b^
	T1	1.869 ± 0.012 ^b^	0.301 ± 0.122 ^a^	4.395 ± 0.016 ^a^	0.472 ± 0.004 ^a^	2.599 ± 0.004 ^b^	0.914 ± 0.004 ^a^	0.073 ± 0.001 ^b^	1.558 ± 0.000 ^a^
	T2	2.161 ± 0.038 ^a^	0.217 ± 0.024 ^a^	4.251 ± 0.031 ^b^	0.403 ± 0.005 ^c^	2.579 ± 0.019 ^b^	0.805 ± 0.006 ^b^	0.077 ± 0.008 ^b^	1.285 ± 0.016 ^c^
41	CK	1.336 ± 0.110 ^b^	0.593 ± 0.018 ^a^	4.325 ± 0.272 ^b^	0.452 ± 0.001 ^c^	5.092 ± 0.239 ^b^	1.408 ± 0.068 ^b^	0.122 ± 0.008 ^a^	0.807 ± 0.054 ^b^
	T1	1.975 ± 0.099 ^a^	0.404 ± 0.121 ^a^	5.673 ± 0.005 ^a^	0.477 ± 0.005 ^b^	5.430 ± 0.017 ^b^	1.336 ± 0.021 ^b^	0.118 ± 0.001 ^a^	0.923 ± 0.000 ^a^
	T2	1.708 ± 0.158 ^a b^	0.353 ± 0.110 ^a^	4.397 ± 0.097 ^b^	0.522 ± 0.004 ^a^	6.165 ± 0.133 ^a^	1.827 ± 0.016 ^a^	0.135 ± 0.001 ^a^	0.957± 0.020 ^a^
48	CK	1.639 ± 0.021 ^a^	0.384 ± 0.018 ^a^	14.042 ± 0.129 ^c^	0.373 ± 0.009 ^b^	6.151 ± 0.077 ^b^	1.722 ± 0.051 ^a^	0.097 ± 0.001 ^a^	0.447 ± 0.006 ^a^
	T1	1.805 ± 0.477 ^a^	0.324 ± 0.065 ^a^	19.331 ± 0.013 ^a^	0.348 ± 0.004 ^b^	6.746 ± 0.037 ^a^	1.419 ± 0.024 ^b^	0.097 ± 0.003 ^a^	0.384 ± 0.006 ^b^
	T2	1.834 ± 0.033 ^a^	0.361 ± 0.065 ^a^	17.452 ± 0.274 ^b^	0.464 ± 0.015 ^a^	6.752 ± 0.232 ^a^	1.482 ± 0.090 ^b^	0.110 ± 0.010 ^a^	0.361 ± 0.007 ^c^
52	CK	1.793 ± 0.095 ^a^	0.411 ± 0.023 ^a^	25.736 ± 0.574 ^b^	0.260 ± 0.007 ^b^	8.590 ± 0.135 ^b^	1.303 ± 0.009 ^b^	0.091 ± 0.000 ^b^	0.454 ± 0.013 ^c^
	T1	1.739 ± 0.039 ^a^	0.421 ± 0.048 ^a^	27.085 ± 0.249 ^ab^	0.197 ± 0.001 ^c^	9.121 ± 0.266 ^ab^	1.426 ± 0.028 ^a^	0.099 ± 0.005 ^ab^	0.713 ± 0.019 ^a^
	T2	1.914 ± 0.007 ^a^	0.357 ± 0.020 ^a^	28.161 ± 0.919 ^a^	0.337 ± 0.005 ^a^	9.238 ± 0.087 ^a^	1.333 ± 0.019 ^b^	0.105 ± 0.003 ^a^	0.540 ± 0.004 ^b^

Note: Results are means ± SE of three independent replications. Different letters represent significant differences between treatments (*p* < 0.05).

## Data Availability

Data is contained within the article or Appendix A.

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
