# Peer review of "Application of Exogenous Melatonin Improves Tomato Fruit Quality by Promoting the Accumulation of Primary and Secondary Metabolites"

_foods, 2022, doi:10.3390/foods11244097_

Round 1

Reviewer 1 Report

The manuscript entitled “Application of exogenous melatonin improves tomato fruit quality by promoting the accumulation of primary and secondary metabolites”, authored by Jianhua Dou , Jie Wang , Zhongqi Tang , Jihua Yu , Yue Wu , Zeci Liu , Junwen Wang , Guangzheng Wang , and Qiang Tian, deals with the investigation of the potential effects derived from the spray application of different concentrations of melatonin (0, 50, and 100 µmol·L-1) on tomato fruits, focusing on changes in primary and secondary metabolites during fruit development. The manuscript contains information that can seriously contribute to knowledge in this field. It appears well written and structured, although several typos are present in the main text. However, I do not feel that this would be a problem. However, some revisions need to be made. Below is a series of comments, listed point by point:

AFFILIATION INFORMATION: In the affiliations section, the email should be listed for each author along with the acronym in brackets. The acronym assigned in this section, should be the same one used later for the contributions section.

ABSTRACT: this section appears to be too unbalanced. Although the concluding section is perfect, little information regarding the current state of the art is present, while too much space is devoted to describing the results. Methodologies are not described. Strongly recommend remodeling this section, even considering that maximum 200 words are allowed.

KEYWORDS: The keywords should be completely changed. The utility of these terms is to facilitate the search of the article using common scientific search engines (PubMed, GoogleScholar, Scopus, etc.), which rely on the terms contained in title, abstract, and keywords. Consequently, using terms that are already in these sections as keywords is inappropriate. I strongly suggest that the keywords be changed before re-submission and add new ones (max 10).

INTRODUCTION: The introduction section has numerous shortcomings. For example,

(i) it begins by talking about tomatoes, but the true object of the authors' study is Melatonin. Consequently, this section should start from line 41. The authors can move line 28-40 to the end of the section, after describing the effects of melatonin on plants. In addition, the authors should explain why they chose to evaluate the effects of melatonin on this very crop.

(ii) Melatonin produced by plant organisms is commonly referred to as 'phytomelatonin,' despite its chemical structure being similar to that of animal origin. This indolamine, in addition to finding useful applications as a biostimulant molecule, covers in plants adept at producing it several physiological functions that the authors should better describe. They can take cues from this recent review that summarizes the effects of melatonin in both animals and plants, marking potential differences (10.3390/ijms22189996).

(iii) The authors should point out that not all plants are able to produce melatonin, and that exogenous application of this molecule as a biostimulant may find useful potential.

(iv) As mentioned earlier, melatonin can be considered a biostimulant molecule for all intents and purposes. The authors should summarily introduce that the use of melatonin in agriculture could be an alternative and sustainable method to the use of agrochemicals. Some information here: 10.3390/agriculture11060557; doi.org/10.1002/jsfa.11318)

MATERIALS AND METHODS: this section is well written, and correctly reports the methods used by the authors. Some suggestions and observations below:

(i) Why did the authors choose to apply melatonin at these concentrations?

(ii) chromatographic analysis should be better described, especially for the detection system, detection and quantification method, and analytical parameters (LOD, LOQ, ME). Please, fix accordingly.

RESULTS: this section is perfect.

DISCUSSION: This section is quite descriptive. The only advice I would give the authors is to divide this section into small subsections. To each subsection, a title highlighting the main effect should be given (i.e. 4.1. Melatonin application increase sugar content in tomato fruits)

CONCLUSION: This section should be implemented. Unfortunately, unlike abstract section, it appears to be too short and lacking in information or achieved data. In addition, the authors should also include potential future perspectives.

Author Response

  Dear Editors and Reviewers:

    Thank you for your letter and for the reviewers’ comments concerning our manuscript. Those comments are all valuable and very helpful for revising and improving our paper, as well as the important guiding significance to our research. We have studied comments carefully and have made correction which we hope meet with approval. Revised portions are marked in yellow in the paper. The main corrections in the paper and the responds to the reviewer’s comments are as flowing:

Reviewer #1:

Point 1: AFFILIATION INFORMATION: In the affiliations section, the email should be listed for each author along with the acronym in brackets. The acronym assigned in this section, should be the same one used later for the contributions section.

Response: Thanks to your valuable suggestions. In the affiliation section, we list the email for each author along with the acronym in brackets and keep it the same as the author contribution section. Added in lines 7 to 10 and added in lines 393 to 397.

Point 2: ABSTRACT: this section appears to be too unbalanced. Although the concluding section is perfect, little information regarding the current state of the art is present, while too much space is devoted to describing the results. Methodologies are not described. Strongly recommend remodeling this section, even considering that maximum 200 words are allowed.

Response: Thanks to your valuable suggestions, we have revised the abstract section. Added in lines 12 to 29.

Point 3: KEYWORDS: The keywords should be completely changed. The utility of these terms is to facilitate the search of the article using common scientific search engines (PubMed, GoogleScholar, Scopus, etc.), which rely on the terms contained in title, abstract, and keywords. Consequently, using terms that are already in these sections as keywords is inappropriate. I strongly suggest that the keywords be changed before re-submission and add new ones (max 10).

Response: Thanks to your valuable suggestions, we have changed the keyword section. Keywords: melatonin; tomato fruit; sugars; organic acids; amino acids; phenolic acids and flavonoids; E-nose; sucrose metabolism. Added in lines 30 to 31.

Point 4: INTRODUCTION: The introduction section has numerous shortcomings.

Response: Thanks to your valuable suggestions. (1) We changed the order of the subsection, describing melatonin and its effects on plants first, followed by tomatoes, and added content. (2) We complemented the differences in melatonin in animals and plants and its physiological functions. Reference: 10.3390/ijms22189996 (3) We also added the useful potential of melatonin as a biostimulant. (4) We also added that the use of melatonin in agriculture could be an alternative and sustainable method to the use of agrochemicals. The following references are cited: 10.3390/agriculture11060557ï¼›doi.org/10.1002/jsfa.11318.

Added in lines 33 to 43 and added in lines 58 to 60 and added in lines 74 to 76.

Point 5: MATERIALS AND METHODS: (1) Why did the authors choose to apply melatonin at these concentrations? (2) Chromatographic analysis should be better described, especially for the detection system, detection and quantification method, and analytical parameters (LOD, LOQ, ME). Please, fix accordingly.

Response: Thanks to your valuable suggestions. (1) We choose to apply melatonin at these concentrations based on previous research results. Our paper has been accepted by journals but has not been published. It can not be cited here. If required, we can provide proof of acceptance. We conclude that 50 µmol·L-1 and 100 µmol·L-1 are the best through principal component analysis. If required, we can provide proof of acceptance. (2) We have supplemented the detection and quantification method and parameters of the testing instrument in the methods. Added in lines 109 to 112 and added in lines 120 to 122 and added in lines 130 to 134 and added in lines 142 to 145.

Point 6: DISCUSSION: This section is quite descriptive. The only advice I would give the authors is to divide this section into small subsections. To each subsection, a title highlighting the main effect should be given (i.e. 4.1. Melatonin application increase sugar content in tomato fruits)

Response: Thanks to your valuable suggestions. To each subsection, we have given a title that highlights the main effect. 4.1 Melatonin application increase sugar content in tomato fruit. 4.2 Melatonin application decrease organic acid content in tomato fruit. 4.3 Melatonin application regulate amino acid content in tomato fruit. 4.4 Melatonin application regulate secondary metabolites content in tomato fruit. 4.5 Melatonin application increase the volatile substance content in tomato fruit.

Point 7: CONCLUSION: This section should be implemented. Unfortunately, unlike abstract section, it appears to be too short and lacking in information or achieved data. In addition, the authors should also include potential future perspectives.

Response: Thanks to your valuable suggestions. We have supplemented this section and made perspectives for the future. Added in lines 389 to 391.

Reviewer 2 Report

Application of exogenous melatonin improves tomato fruit quality by promoting the accumulation of primary and secondary metabolites by Jianhua et al., studied various factors to improve tomato fruit quality through application of metabolites. The study will facilitate breeders and also help in yield of tomato fruit. However, there are some deficiencies which must be addressed.

Provide quantitative results in the abstract section.

Also add one to two future recommendations in the abstract.

Replace word fruits with fruit in the whole MS. As plural of fruit is fruit.

Line 39 add genetic breeding and Phytostimulatory activities by citing the following relevant studies

https://doi.org/10.1007/s10725-021-00785-7, DOI: 10.1016/j.micpath.2020.103966,

Line 50 should be cited with relevant study.

https://doi.org/10.3390/genes13101699,

Line 62-63 which previous studies?

Line 65 mention references of the few reports.

Add commercial importance of melatonin.

Section 2.2 and 2.3 provide complete methods.

Conclusion provide future recommendations of the study.

Author Response

  Dear Editors and Reviewers:

    Thank you for your letter and for the reviewers’ comments concerning our manuscript. Those comments are all valuable and very helpful for revising and improving our paper, as well as the important guiding significance to our research. We have studied comments carefully and have made correction which we hope meet with approval. Revised portions are marked in yellow in the paper. The main corrections in the paper and the responds to the reviewer’s comments are as flowing:

Reviewer #2:

Point 1: Provide quantitative results in the abstract section.

Response: Thanks to your valuable suggestions. We have revised the abstract section. However, considering that we measured the indicators as dynamic data and the indicators are all components, providing quantitative results for each indicator would lead to a longer and non-compliant abstract, which we hope can be understood. Added in lines 12 to 29.

Point 2: Also add one to two future recommendations in the abstract.

Response: Thanks to your valuable suggestions. we have added future recommendations in the abstract. Added in lines 27 to 29.

Point 3: Replace word fruits with fruit in the whole MS. As plural of fruit is fruit.

Response: Thanks to your valuable suggestions, we have replaced the word fruits with fruit in the whole MS.

Point 4: Line 39 add genetic breeding and Phytostimulatory activities by citing the following relevant studies.

Response: Thanks to your valuable suggestions, we have added genetic breeding and Phytostimulatory activities by citing the following relevant studies. Added in lines 39 to 40. And cite the following references:

https://doi.org/10.1007/s10725-021-00785-7, doi: 10.1016/j.micpath.2020.103966

Point 5: Line 50 should be cited with relevant study.

Response: Thanks to your valuable suggestions, we have cited relevant study reports in lines 49. For example: https://doi.org/10.3390/genes13101699.

Point 6: Line 62-63 which previous studies?

Response: Thanks to your valuable suggestions, we have provided an account of previous research and current less studied areas in lines 74 to 76. Previous studies on melatonin have focused on stress resistance of plants, or postharvest quality and preservation of fruit. However, there are few reports on the effects of melatonin on primary and secondary metabolites during fruit growth and development.

Point 7: Line 65 mention references of the few reports.

Response: Thanks to your valuable suggestions, we have added the following references of the few reports. Added in lines 80. And cite the following references:

doi: 10.1093/jxb/ert443; doi: 10.1016/j.scienta.2022.111008

Point 8: Add commercial importance of melatonin.

Response: Thanks to your valuable suggestions, we have increased the commercial importance of melatonin. Added in lines 58 to 60.

Point 9: Section 2.2 and 2.3 provide complete methods.

Response: Thanks to your valuable suggestions, we have provided the complete methods in Sections 2.2 and 2.3. Sections 2.2: added in lines 109 to 112. Sections 2.3: added in lines 120 to 122.

Point 10: Conclusion provide future recommendations of the study.

Response: Thanks to your valuable suggestions, future recommendations for this study have been added in conclusion. Added in lines 389 to 391.
